# A Vision-Based System for Stage Classification of Parkinsonian Gait Using Machine Learning and Synthetic Data

**DOI:** 10.3390/s22124463

**Published:** 2022-06-13

**Authors:** Jorge Marquez Chavez, Wei Tang

**Affiliations:** 1Department of Physics, New Mexico State University, Las Cruces, NM 88003, USA; jorgemar@nmsu.edu; 2Klipsch School of Electrical Engineering, New Mexico State University, Las Cruces, NM 88003, USA

**Keywords:** Parkinson’s disease, gait analysis, vision-based system

## Abstract

Parkinson’s disease is characterized by abnormal gait, which worsens as the condition progresses. Although several methods have been able to classify this feature through pose-estimation algorithms and machine-learning classifiers, few studies have been able to analyze its progression to perform stage classification of the disease. Moreover, despite the increasing popularity of these systems for gait analysis, the amount of available gait-related data can often be limited, thereby, hindering the progress of the implementation of this technology in the medical field. As such, creating a quantitative prognosis method that can identify the severity levels of a Parkinsonian gait with little data could help facilitate the study of the Parkinsonian gait for rehabilitation. In this contribution, we propose a vision-based system to analyze the Parkinsonian gait at various stages using linear interpolation of Parkinsonian gait models. We present a comparison between the performance of a k-nearest neighbors algorithm (KNN), support-vector machine (SVM) and gradient boosting (GB) algorithms in classifying well-established gait features. Our results show that the proposed system achieved 96–99% accuracy in evaluating the prognosis of Parkinsonian gaits.

## 1. Introduction

Parkinson’s disease (PD) is a neurodegenerative disease that is characterized by motor symptoms, such as tremor, rigidity and bradykinesia [1]. These movement impairments are directly linked to a variety of abnormal gait patterns in PD patients, which can subsequently increase the risk of injury and affect the overall quality of life [2]. The current gold standard for the diagnosis and monitoring of the Parkinsonian gait (PG) generally consists of clinical evaluation. However, the criteria used is often based on the examiner’s expertise, and the implementation can therefore attach unwanted subjective components to the analysis [3]. Moreover, since PD is a progressive condition, the variation introduced by human interpretation makes clinical-based evaluations further unsuitable for the early detection of PG.

To avoid the inaccuracy of these methods, several technological advances have made it possible to perform gait analysis through specialized equipment. Previous studies, for example, have consisted on the use of walkways or wearable sensors [4,5] to obtain important features and perform gait analysis [6,7]. Still, although these techniques provide a rich quantitative examination that can produce relevant data not observed by the eye, they are generally considered impractical, as they often require costly additional equipment that can be inconvenient to both the examiners and the patient [2,8,9].

Furthermore, the results obtained with similar body-worn technology were shown to be affected by the patients changing their walking pattern upon acknowledging the sensors themselves, leading to inaccurate results [10]. In this context, the development of a vision-based system has gained increased attention because of its ability to objectively quantify gait features through a practical camera setup and machine-learning algorithms.

Previous work has shown the use of both marker [11,12] and markerless [13,14,15] pose-estimation methods to differentiate the Parkinsonian gait from normal and other abnormal types of gait. To account for the worsening of PD symptoms over the course of the disease, other studies have introduced methods for stage classification of PD based on gait analysis [16,17,18]. This work is useful because it can determine how specific gait features affect the worsening of the disease, and it can be used to implement a continuous adjustment to therapy [19].

Nevertheless, although these studies have been able to quantify severity levels of PD according to the patients’ gait, their reproducibility is often limited by the restrictions applied to the data used, as well as by the impracticality introduced by the sensor-based methodologies [17,20]. This lack of access to large data sets exemplifies a significant challenge in the study of vision-based systems in the clinical field, as it not only restricts the analysis of PG to only the specific available examples but also can lead to inaccurate conclusions obtained from possibly overfitted results [21].

In machine learning, an alternative to the acquisition of real data is the generation of synthetic data, which attempts to preserve the properties of the original data set while allowing for more information to be fed into the developed model [22]. There are different methods to obtain this data, such as adding random noise to existing data points or creating a mathematical model of the available real information to simulate more data points [23]. The main goal of this approach is that the artificially manufactured information can be used to expand on existing real data and train deep-learning models [24]. As such, the generation of synthetic data could be a valuable tool to overcome the lack of access to real data in health care [25].

In this paper, a vision-based system that performs stage classification of Parkinson’s disease is proposed (Figure 1). We use a pose-estimation software to extract gait features from recorded videos with the goal of assigning the obtained data to a specific severity level of the disease through the training of three machine-learning classifiers: k-nearest neighbors (KNN), support-vector machine (SVM) and gradient boosting (GB). Synthetic data was used to train the models, and this was generated by performing a linear combination obtained from a normal and a Parkinsonian gait to simulate examples of PD at multiple severity levels.

Each model was hyper-parametrized and verified by evaluating its respective learning curve. This analysis, combined with the performance of metrics for each classifier, led to three trained models with an accuracy of 96–99%. An additional video that was independent from the original training of the classifiers showing a subject with early Parkinson’s disease is tested on one of the models, and the stage predicted is consistent with the characteristics of the patient.

## 2. Materials and Methods

### 2.1. Skeleton Extraction and Feature Analysis

Our methodology is summarized in Figure 2. The motion of the patient was first recorded using a stationary camera. This input was then fed into the OpenPose system [27], which returned a combination of two dimensional coordinates that specify the location of up to 32 human body parts for each frame; the OpenCV library [28] was used to draw the located body parts on each video as shown in Figure 3. Although this marker-less system does not provide the exact position of each body part, previous work demonstrated that OpenPose-based motion capture has an accuracy of 30 mm or less [29].

Additionally, marker-based sensors themselves, despite being more typically used, also have intrinsic errors due to deformation of the skin on which they are attached [29]. Hence, this error was considered small for our purposes.

The behavior of the set of coordinates of the heel, ankle and hip markers were graphed as a function of time, and two data segmentation types were created to subsequently obtain the gait features: Type I (T1) segmentation demonstrates the distance between each ankle, while Type II (T2) segmentation shows the y-coordinate of each foot obtained from the pose-estimation system (Figure 4). In both data segmentation types, the amplitudes were normalized according to the maximum value of the dependent variable obtained from the pose estimation algorithm.

From these signals, the following commonly used features to analyze gait were obtained: step length, step time, stride time, stride length, swing time and double support. From Type I segmentation, stride time was obtained by taking the difference in the time axis between every other extremum, while the stride length was derived by adding subsequent maxima/minima; step time was obtained by taking the difference in the time axis between every extremum, while the step length was defined as the extrema (i.e., when the distance between each foot reaches its maximum value at the ground).

From Type II segmentation, the double support feature was obtained by taking the difference in the time axis at the minima of each foot, while swing time was obtained by the same difference of a single foot. The scipy peak detection algorithm [30] was used to extract the maxima and minima of each curve and the signals were denoised using the fast Fourier transform (FFT) algorithm from the numpy library [31]. Table 1 gives a summary of each feature and the method with which it was derived in its corresponding type of data segmentation.

### 2.2. Selecting Synthetic Dataset and Hyper-Parameters

#### 2.2.1. Data Generation

Two videos were obtained to construct the data set employed: a Youtube video [32] was used as a reference for Parkinsonian gait while a video recording of a subject walking on a treadmill was used for normal gait. Both videos were recorded from the patient’s side. The synthetic data generated was obtained by using a linear combination of the each data segmentation curve from both normal and Parkinsonian gait to simulate additional examples at different severity levels.

As the time at which the OpenCV algorithm detects each body part varies according to each video, the curve of each data segmentation type was interpolated for a selected period of time in which the pose-estimation model made detections for both normal and Parkinsonian gait; this created a continuous set of values and permitted the combination of both gaits. For each data point at a specific time, the linear combination $l_{C}$ was calculated (Figure 5) according to the equation,
(1)$$l_{C}=l_{P}\omega +\left(1-\omega \right)l_{N}$$
where *l*${}_{P}$ refers to the data value from PG, *l*${}_{N}$ to that from normal gait and ω is a weight factor that ranges between 0 and 1.

To test all three classifiers, three data sets were generated using Equation (Equation 1). The following number of linear combinations, corresponding to the number of ω factors, were used to create such data sets: [20, 40, 60, 80, 100, 200, 250]. Each model was tested with the generated data sets, and only the ones that showed no overfitting or underfitting were selected for each model.

The ω factor created a specific set of curves, representing a synthetic gait, from which all six features were obtained and labeled according to the value of such factor. Four different stages (I, II, III, and IV) representing the severity level of the disease (healthy, mild, moderate and severe) were defined based on the Hohen and Yahr (H&Y) scale, similar to the classification process used in [17]. Level I was defined for 0.0 ≤ω< 0.25, level II for 0.25 ≤ω< 0.50, level III for 0.50 ≤ω< 0.75 and level IV for 0.75 ≤ω≤ 1.0. When ω = 1, the data was defined to correspond to the curve obtained from PG and when ω = 0, $I_{C}$ was set to correspond to the curve obtained from thre normal gait.

#### 2.2.2. Classification Models

The KNN, SVM and GB models were trained and evaluated using 10-fold (stratified) cross-validation to help reduce overfitting effects. For each classification algorithm, the corresponding hyper-parameters were optimized by employing a grid search strategy. Table 2 shows these characteristics for each classification model.

#### 2.2.3. Learning Curves to Test Quality of Fit

Once the data was generated, it was used to train a machine-learning model to assign a severity level to PG. However, given the definition of ω, the number of linear combinations could have increased or reduced the size of the data set and therefore directly impacted the quality of the training of the model. As such, for every combination of hyper-parameters in each classification model, a learning curve (LC) was obtained to first verify the validity of the results.

LCs are valuable tools that can show the measure of the predictive accuracy on the test examples as a function of the number of training examples [33], and they represent the performance of a model in terms of the size of the training set such that their analysis can help determine if the model is overfitted or underfitted [34]. Overfitting happens when the predicted values match the actual values in the training set too well [35]. As a result, the model now learns the training data to such a high degree that it cannot detect unseen data sets well [36]. In contrast, underfitting occurs when the simplicity of the model translates into a poor representation of the population data [37].

The learning curve of a good fit model has high training loss at small training size and it gradually decreases as more training examples are added; when the training size is large enough, the curve appears to flatten gradually such that adding more training examples will not improve the model performance. Similarly, the validation loss values are very high at small training data sets; however, they begin to decrease and eventually flatten out as more training examples are added [38].

The learning curves of all selected hyper-parametrization combinations were obtained using the sklearn Python library [39] and assessed using the previously mentioned features. The developed code selected only the graphs that showed the characteristics of a good fit of the learning curve.

### 2.3. Performance Evaluation

Once the fit was determined, the classification models with its corresponding hyper-parameters were evaluated with accuracy (the ratio of correct prediction to all observations), sensitivity (the ratio of correct prediction of positive cases to positive items), specificity (characterizes negative probability of true prediction to true negatives), F1 score (the harmonic mean of precision and sensitivity), PPV or precision (ratio between positive predictions to real positives) and NPV (ratio of negative predictions to true negatives). These were obtained with the following Equations (TP = true positive, TN = true negative, FP = false positive and FN = false negative):(2)$$Accuracy=\frac{TP+TN}{TP+FN+TN+FP}$$
(3)$$Sensitivity=\frac{TP}{TP+FN}$$
(4)$$Specificity=\frac{TN}{TN+FP}$$
(5)$$F1score=2*\frac{Precision*Sensitivity}{Precision+Sensitivity}$$
(6)$$PPV=\frac{TP}{TP+FP}$$
(7)$$NPV=\frac{TN}{TN+FN}$$

Furthermore, a Receiver Operating Characteristic (ROC) curve was obtained to show the trade-off between the sensitivity and specificity. As this type of graph is intended for the analysis of binary classification problems, each label was considered one at a time with all the other groups as one. This graph indicates how effectively the classifier can distinguish each severity level; the greater the AUC (Area Under the Curve), the better the accuracy of the model [40].

### 2.4. Test Data on Trained Models

After the validity of each model was verified with both the learning curve and the performance evaluation metrics, an additional video of a subject with early Parkinson’s disease was obtained from Youtube to predict the stage of the patient [41]. The video shows a patient with a recent onset of Parkinson’s disease. The information obtained from this additional sample was then input into the previously trained model that showed the best performance metrics and provided the best fit.

The motivation behind this procedure was to test the developed model with an independent data set that had not been part of the synthetic data generation nor the training process and, thus, provide an unbiased evaluation of the classifier fit on the training data set.

The video obtained was analyzed using the pose-estimation procedure and the same method for feature extraction was carried out. In this case, the synthetic data was not generated again. The results showed 19 data points for all features used. This data set was then input into the previously trained model that showed the best performance to predict the classification stage for each feature. The expected result was a severity level of II or III, given the description of early PD from the video.

## 3. Results

The numbers of ω factors selected were 40 for the GB model, 80 for the KNN model and 200 for the SVM model. As such, the number of features in each data set varied from model to model. For the KNN classifier, 724 features were identified; for the GB model, 362 features were used, and there were 1830 features for the SVM model. These were plotted according to their stage classification in Figure 6.

For the KNN model, the biggest percentage difference between the majority and the minority class was 10.4%; this value was slightly less for the SVM model (8.8%) and greater for the GB model (12.7%). Given that this class imbalance was not very large, no modules, such as the SMOTE (Synthetic Minority Over-sampling Technique) [42] were used to create more samples for the minority classes in each fold for the training data.

From all the combinations of hyper-parameters selected with the different data sets generated, the code only identified one curve for each model that showed a good fit behavior (Figure 7). These hyper-parameters, as well as the selected number of linear combinations that created each data set, are specified in Table 3. Although the GB model required the least number of linear combinations (40) for the training loss and validation loss curve to stabilize—in comparison to the KNN model (80) and SVM (200) models—the difference in the value to which each curve converged to was greater for this model.

This difference between the training and validation loss can be seen when comparing their average accuracies. For example, the difference between the training accuracy (97.2%) and the average of the cross-validation accuracies (93.4%) was 3.8% for the GB classifier, which was greater than both the KNN model (with a 95.4% training accuracy and an average of the cross-validation accuracies of 94.6%) and the SVM model (with a 88.5% training accuracy and an average of the cross-validation accuracies of 91.3%).

Accordingly, the standard deviation of the GB classifier was also greater with a value of 0.03%, relative to KNN (0.02%) and SVM (0.014%). These features show that some overfitting might have occurred with the data set chosen to test the trained GB model. Still, for all three curves, their general behavior was congruent with the characteristics of a good fit and both the training and validation loss curves eventually converged to a similar value. Moreover, the area under the ROC curve was close to or equal to one in all of the classes for each classifier, showing that the true positive rate was greater than the false-positive rate (Figure 8).

Table 4 presents the performance metrics for each severity level, and these are illustrated in Figure 9. The confusion matrices in Figure 10 also show the performance of each model at classifying every severity level with a normalized version of the predicted and true results. The cumulative performance of the classifiers are included in Table 5. The GB model showed the greatest overall accuracy with a value of 0.99, followed by KNN (0.97) and SVM (0.96). Given that SVM has the lowest NPV, it shows the lowest misclassification rate for all classes.

The high accuracy of the GB model can be explained by the quality of the fit of the learning curve, as the training and cross-validation loss difference was greater for this classifier than for the other two models. To account for the slight imbalanced data, the F1 score was also considered, as this is a more adequate tool to measure these type of data sets given that it gives more weight to classified samples in the minority class [43]. According to the average F1 score of each class across all three classifiers, the severity levels I (0.96) and IV (0.97) appear to have the highest values, in contrast to levels II (0.91) and III (0.92).

Figure 11 shows the contribution of each of the gait features in the classification model. Both the KNN model and SVM model showed very similar values for each feature, with the swing time of each foot, double support and step length being among the most important factors in each classifier. For the GB model, the swing time, double support, stride length and step length were the most important features.

The positive contributions of each feature aligns with other studies; the stride length, step length and double support time were found to be significantly different between PD patients and controls [44], while the swing time was suggested as a predictive variable for PD [45]. The difference in the contributions of certain features between the GB and the SVM/KNN model can be attributed to the slight overfit behavior shown by the GB learning curve.

The KNN model was the classifier selected to test the unbiased data set, given that its learning curve mimicked a good fit behavior by showing the lowest difference between the cross-validation and training accuracy while still having high performance evaluation measurements.

The stage predictions for each set of features (step length, step time, stride time, stride length, swing time and double support) made by this model showed a high number of set of features classifying the patient’s gait as level II (12), with a low number of features classifying it as I (2) and IV (5). Given the description of the early stage of the subject in the video selected, these results are congruent with the severity of gait shown by the patient.

## 4. Discussion

This study aimed to distinguish different severity levels of Parkinsonian gait by using machine-learning models and analyzing gait features obtained from a pose-estimation algorithm. Several studies have proposed a similar methodology to perform a stage classification of PG. For example, in [20], a vision-based system was introduced to estimate the severity of PG and compare the classification results to the Unified PD Rating Scale (UPDRS) ratings.

Although there was a strong correlation between the clinician labels and the model estimates, the classifiers implemented were trained using data that had availability restrictions; such limitations are an obstacle to other studies that have had to rely on significantly smaller data sets to carry out their analysis. For instance, in [46], the study consisted of 37 patients, while in [47], the proposed system used the data from 23 patients with Parkinson’s disease, a number comparable to that of participants in [48].

Similarly, additional studies [17,49] have resorted to the use of the public data set for gait pattern provided by Physionet [50], which contains data of 93 PD patients obtained from 16 sensors measuring the ground reaction force of the patients as a function of time. The accuracy of the proposed method is compared to other studies that have used machine-learning algorithms to classify PG in Table 6. This paper provides an alternative to the these methods by creating additional information based on real, pre-existing data and implementing it into a practical pose-estimation algorithm.

Although a significant amount of work remains for classification stages through deep-learning technologies to be used in the clinical field on a regular basis, our process of synthetic generation is a useful tool that can satisfy the need for acquiring larger data sets suggested by previous studies [21,30,48,52] through sensor-less technology.

### 4.1. Generation of Synthetic Data

Synthetic data was generated from the analysis of a Youtube video showing PG and normal gait of a subject walking on a treadmill. We were able to create an optimal data set using a linear combination of these type of gait and selected the hyper-parameters that produced the best fit by a given machine-learning classifier. The creation of an artificial point from a linear combination of different data sets is a technique that has been previously implemented in machine learning across other fields.

For instance, in [53], they generate a synthetic data set by taking a linear combination of bus-level load patterns found from real data due to the limited complexity of the load profiles available, and a similar idea is applied in forecasting, where a linear combination of factors is used to improve the model’s accuracy [54].

The use of synthetic data has also been applied in the biomedical field, where augmented data has been developed to increase the number and variability of examples [55]. Furthermore, different studies with a similar goal as this paper’s have also focused on generating synthetic data to incorporate it to the training set and avoid class imbalance problems [56,57,58]. However, to the best of our knowledge, no other studies have implemented the use of synthetic data to simulate Parkinsonian gait and generate independent data sets.

### 4.2. Result Interpretation

By analyzing the model with a learning curve, we were able to validate the quality of our results and verify that the model would not create perfect evaluation metrics only due to the high amount of data that could be generated with the proposed method.

The performance evaluation of the chosen classifiers showed that the KNN model had a good fit with its training and average cross-validation accuracy being no more different than 0.8%, as well as having a high accuracy of 0.97. A good fit learning curve was also produced by the SVM model with similar metric performance. For the GB model, however, although the model showed a high accuracy and F1 score, the learning curve produced shows potential overfitting given the difference between the cross-validation and training accuracy.

Accordingly, the GB model is the classifier whose feature contribution deviates the most from all three models. Nevertheless, the feature analysis of all models reveals a positive contribution from all features and gives a high importance to the swing time, step length and double support, results that are congruent with previously found characteristics that can predict PG.

### 4.3. Practical Applications

PD is a neurodegenerative disease whose progression is gradual such that patients show different motor symptoms at various severity levels and require different means of treatment. For example, in the early stages of PD, a small dose of oral levodopa (L-DOPA) [59] is a common effective treatment. As such, assessing the stage of the disease can help improve the management of patients. Our approach can also help clinicians by providing them with features that can lead to individualized treatments.

In the advanced stages of the disease, for example, the use of deep brain stimulation (DBS) can benefit from adjustments according to these features [20]. The development of technologies that quantify Parkinsonian gait can therefore aid clinicians in making adequate beneficial changes in patient treatments.

Despite the advancement of technologies that have aimed to accomplish this task, gait analysis has not been widely used by clinicians because of cost and practicality [60]. In contrast to the current PD rating based on walkways and wireless wearable sensors [61,62]—which are limited by battery lifetime [63] or expensive lab environment—the proposed method is less time-consuming, avoids errors due to inconsistent marker placement and does not require additional equipment.

Moreover, it avoids potential subjective bias introduced by medical scales by not relying on medical experts with specialized knowledge. The implementation of synthetic data helps avoid the current challenges of having limited access to data of the Parkinsonian gait. In this way, given that our methodology also does not depend on any specific recording device, similar vision-based systems could be used for assessing patients using an application in any device.

### 4.4. Limitations

The current contribution has several limitations. First, the accuracy of the marker-less approach would be less than the market-based approach. However, the primary goal of this work is to try to provide convenient assistance to the medical practitioner to help them evaluate the stages of Parkinson’s disease through gait analysis using video recordings. The proposed system will not replace human medical practitioners, and thus it would not require very high accuracy since it is not directly making medication decisions.

The proposed approach is only an assistive technology. The primary advantage of this work is applying synthetic data in machine learning to reduce the cost of data and label creation in training machine-learning models. Moreover, the vision-based method does not require expensive motion sensing systems, which provides convenience for the patient and the medical practitioners to recognize the stage of the patient and help the medical practitioners make a decision.

In our study, the initial data of both Parkinsonian and normal gait from which the linear combinations are generated can be significantly increased to produce more oscillations in the segmentation types and therefore produce more accurate measurements. The study also does not include other gait features that have been established as important characteristics to identify abnormal patterns in Parkinsonian gait, such as freeze of gait (FOG), step width, step velocity and stance time asymmetry [64].

Our approach also focused on selecting the portion of the video when the patient would start or stop walking, and no sections of the videos where the subjects are turning were analyzed. Furthermore, the videos obtained were only recorded from one side of the patient, which could have affected the pose-estimation system’s ability to identify the side of the body not facing the camera directly. Future implementations of a similar methodology should include a variation of angle in the video recordings to augment the acquisition of gait features.

Additionally, it is important to note that the creation of any synthetic data set must be congruent with the nature and characteristics of the original data set [65]. Thus, this aspect of using synthetic data should be heavily analyzed. More complex algorithms that encompass additional characteristics of the subject, such as weight, height, or sex could be incorporated to minimize the loss of information we proposed through our linear combination method, although as pointed out in [17], the spatiotemporal variables obtained from our analysis are less likely to be affected by these physiological parameters.

## 5. Conclusions

This paper presented a vision-based gait classification system to diagnose the severity of Parkinson’s disease based on the analysis of gait features through synthetic data. Three machine-learning models, including KNN, SVM and GB algorithms, were applied, and their performances were compared. The system achieved acceptable accuracy using synthetic data. To the best of our knowledge, this work is the first to apply synthetic data to simulate a Parkinsonian gait and to generate independent data sets. This method could help provide quantitative results of diagnosis and evaluations of the rehabilitation process using limited data, which is an attractive method for implementing future clinical anomaly motion analysis.

## Figures and Tables

**Figure 1 sensors-22-04463-f001:**
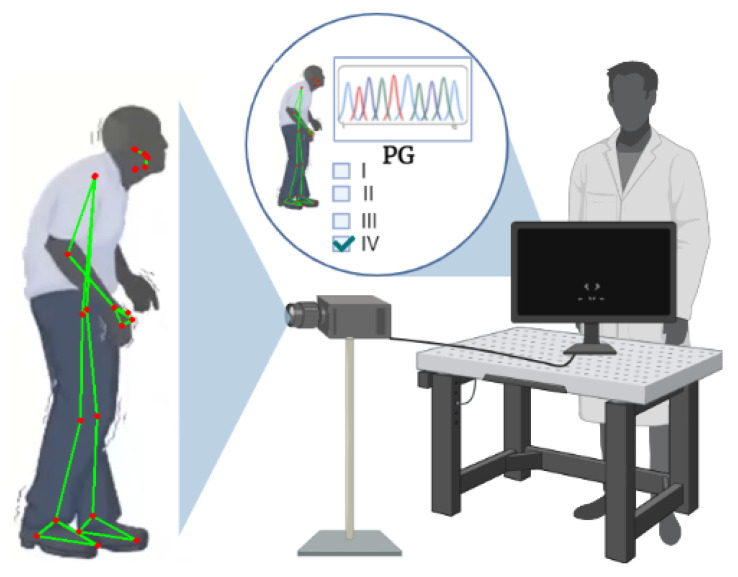
Vision-based system for stage classification of Parkinson’s disease based on gait analysis. Created with [26].

**Figure 2 sensors-22-04463-f002:**
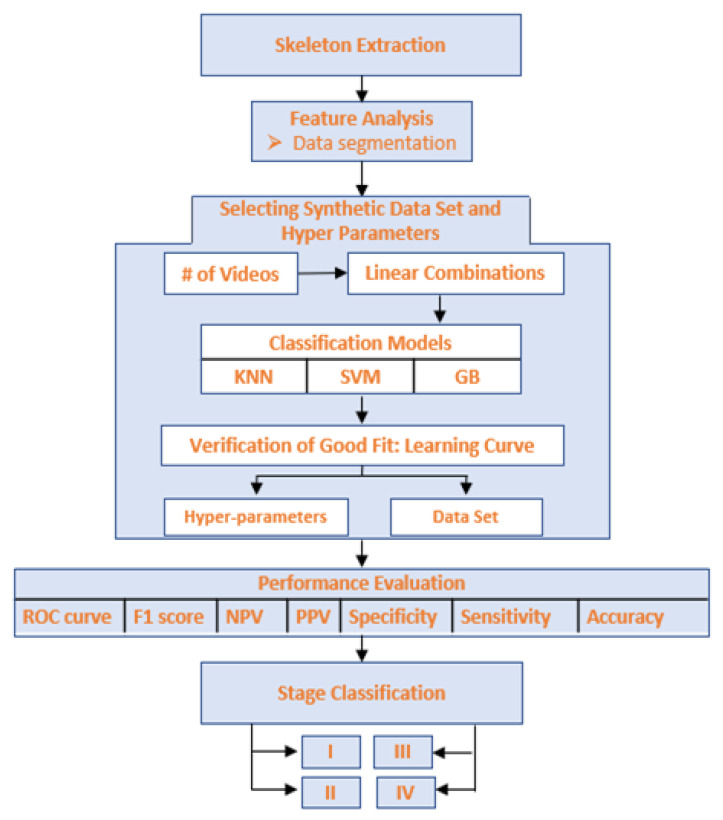
Proposed method for Parkinsonian gait analysis using pose estimation and synthetic data.

**Figure 3 sensors-22-04463-f003:**
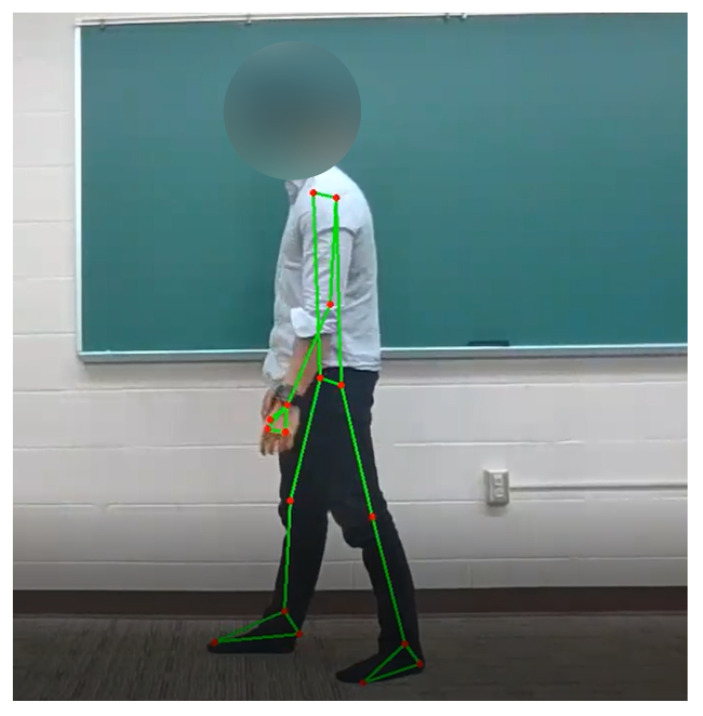
Example of pose estimation using human pose data.

**Figure 4 sensors-22-04463-f004:**
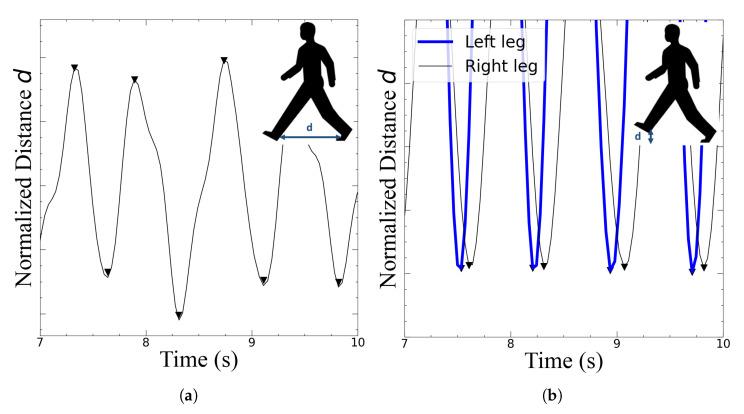
(**a**) Type I and (**b**) Type II data segmentation types.

**Figure 5 sensors-22-04463-f005:**
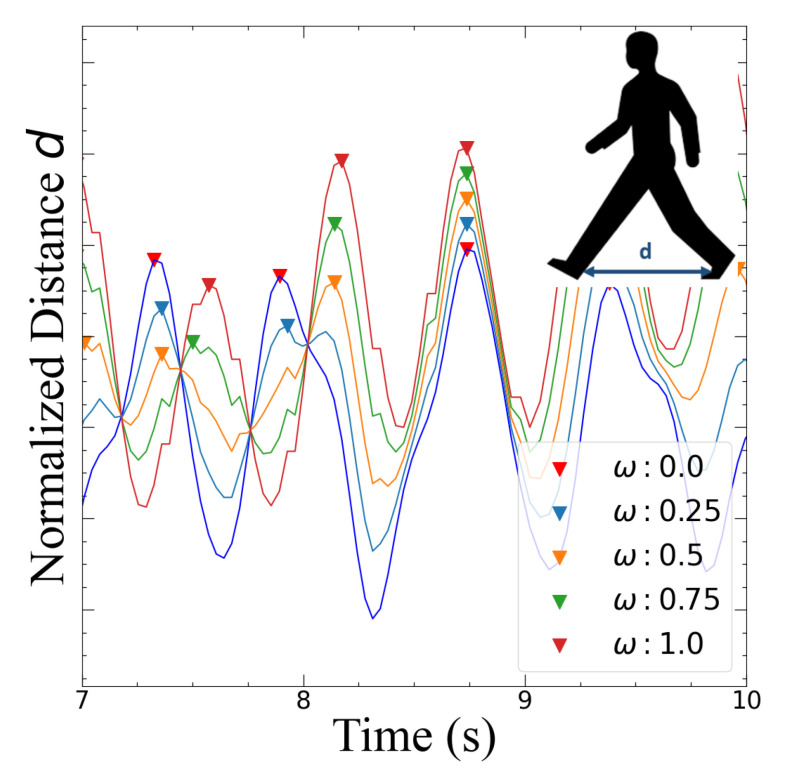
Example of generation of synthetic data using a linear combination of a Parkinsonian and normal gait.

**Figure 6 sensors-22-04463-f006:**
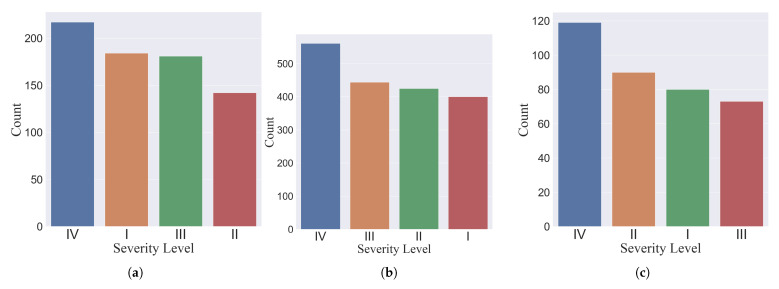
Distribution of data set according to each class for the (**a**) KNN, (**b**) SVM and (**c**) GB models.

**Figure 7 sensors-22-04463-f007:**
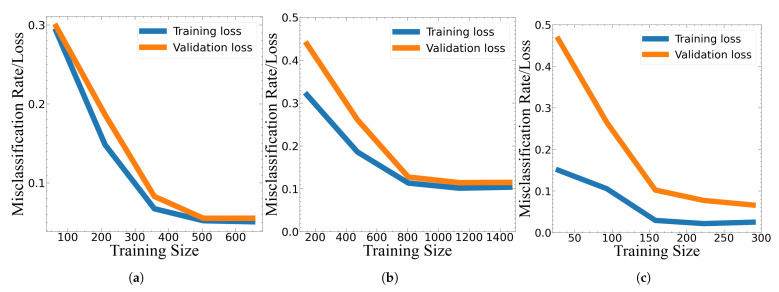
Learning curves for the (**a**) KNN, (**b**) SVM and (**c**) GB models.

**Figure 8 sensors-22-04463-f008:**
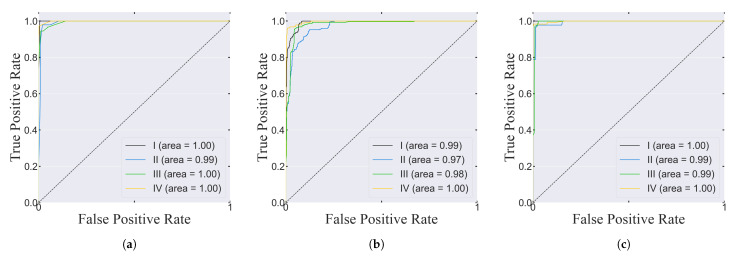
ROC curves for the (**a**) KNN, (**b**) SVM and (**c**) GB models.

**Figure 9 sensors-22-04463-f009:**
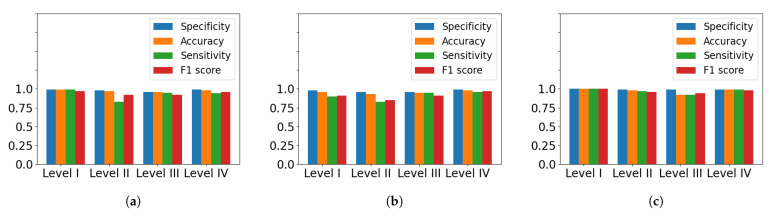
Visualization of the general performance metrics for the (**a**) KNN, (**b**) SVM and (**c**) GB models.

**Figure 10 sensors-22-04463-f010:**
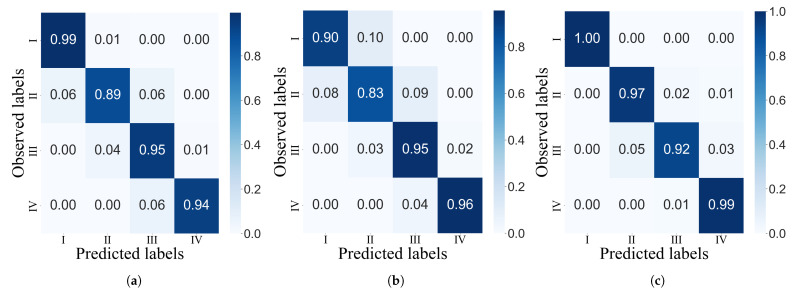
Confusion matrices for the (**a**) KNN, (**b**) SVM and (**c**) GB models.

**Figure 11 sensors-22-04463-f011:**
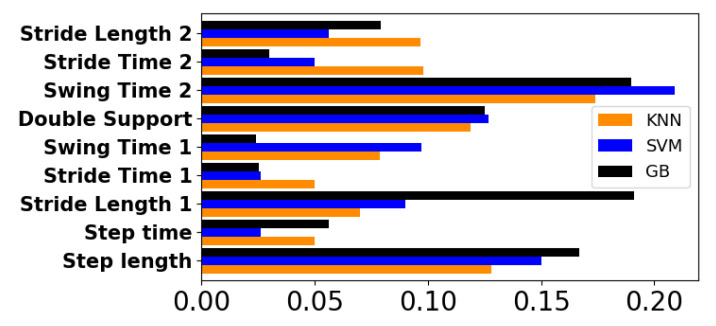
Feature importance of the KNN, SVM and GB models. For some features, the number assignation refers to a single foot.

**Table 1 sensors-22-04463-t001:** Data segmentation types for feature extraction.

Feature	Segmentation	Description	Calculation
Stride time	Type I	Duration between the first contact of the two consecutive footesteps of the same foot	t${}_{i+2}$ − t${}_{i}$ where $extremum[T1\left(t_{i}\right)]$ for *i* = 1, 2, 3...
Stride length	Type I	Distance between successive points of contact of the same foot	$extremum[T1\left(t_{i}\right)]$ + $extremum[T1\left(t_{i+1}\right)]$ for *i* = 1, 2, 3...
Step time	Type I	Duration between consecutive heel strikes	t${}_{i+1}$ − t${}_{i}$ where $extremum[T1\left(t_{i}\right)]$ for *i* = 1, 2, 3...
Step length	Type I	Distance between the contact of one foot and contact of the opposite	$extremum[T1\left(t_{i}\right)]$ for *i* = 1, 2, 3
Double Support	Type II	Period in which both feet are in contact with the floor	t${}_{i}$ − t${}_{j}$ where $min{\left[T2\left(t_{i}\right)\right]}_{foot1}$ and $min{\left[T2\left(t_{j}\right)\right]}_{foot2}$ for *i* = *j* = 1, 2, 3...
Swing time	Type II	Period in which only one foot is the ground	t${}_{i+1}$ − t${}_{i}$ where $min{\left[T2\left(t_{i}\right)\right]}_{foot}$ − $min{\left[T2\left(t_{i}\right)\right]}_{foot}$ for *i* = 1, 2, 3...

**Table 2 sensors-22-04463-t002:** The hyper-parameters tested for each model.

Classification Models	Hyper-Parameters Selected
KNN	n_neighbors = [1, 3, 5, 7, 9, 11, 13, 15], metric = [euclidean, manhattan, minkowski], weights = [uniform, distance]
SVM	C = [0.01, 0.1, 1, 10, 100], kernel = [linear, rbf], gamma = [scale]
GB	n_estimators = [1, 2, 5, 20, 50, 100]

**Table 3 sensors-22-04463-t003:** The hyper-parameters obtained using a specific number of videos for the (a) KNN (b) SVM and (c) GB models.

Model	KNN	SVM	GB
Hyper-parameters	metric: euclidean, n_neighbors: 15, weights: uniform	C: 0.1, gamma: scale, kernel: rbf	n_estimators: 2
Number of linear combinations	80	200	40

**Table 4 sensors-22-04463-t004:** Performance metrics per class of each model tested.

Model	Severity Level	TN	TP	FN	FP	Accuracy	Sensitivity	Specificity	F1 Score
KNN	I	532	183	1	8	0.99	0.99	0.99	0.98
	II	573	126	16	9	0.97	0.83	0.98	0.91
	III	521	172	9	22	0.96	0.95	0.96	0.92
	IV	506	203	14	1	0.98	0.94	0.99	0.96
SVM	I	1396	359	41	34	0.96	0.90	0.98	0.91
	II	1349	354	71	56	0.93	0.83	0.96	0.85
	III	1324	421	23	62	0.95	0.95	0.96	0.91
	IV	1261	536	25	8	0.98	0.96	0.99	0.97
GB	I	282	80	0	0	1	1	1	1
	II	268	87	3	4	0.98	0.97	0.99	0.96
	III	286	67	6	3	0.98	0.92	0.99	0.94
	IV	240	118	1	3	0.99	0.99	0.99	0.98

**Table 5 sensors-22-04463-t005:** Hyper-parameters obtained using a specific number of videos for the (**a**) KNN, (**b**) SVM and (**c**) GB models.

Performance Evaluation Measurements	KNN	SVM	GB
Sensitivity	0.94	0.91	0.97
Specificity	0.98	0.97	0.99
Accuracy	0.97	0.96	0.99
PPV	0.95	0.91	0.97
NPV	0.98	0.97	0.99
F1 score	0.94	0.90	0.97

**Table 6 sensors-22-04463-t006:** Comparison of accuracies obtained with other methodologies.

References	Data acquisition	Methods	Accuracy
Rupprechter et al. [20]	729 subjects	RFC, LDA, LOGIS, ANN, SVM, XGBoost	47.0–50.0%
Balaji et al. [17]	Physionet	DT, SVM, EC, BC	69.7–99.4%
Abdulhay et al. [49]	Physionet	Medium Tree, Medium Gaussian SVM	90.0–94.8%
Aich et al. [51]	20 PD subjects	RF, SVM, KNN, Naïve Bayes	86.0–96.7%
Proposed method	Synthetic data	SVM, GB, KNN	96.0–99.0%

## Data Availability

The data presented in this study are available on request from the corresponding author.

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
