# Peer review of "A Vision-Based System for Stage Classification of Parkinsonian Gait Using Machine Learning and Synthetic Data"

_sensors, 2022, doi:10.3390/s22124463_

Round 1

Reviewer 1 Report

This study proposes a visualization system to collect the gait of the patients with Parkinson's disease and analyze the disease severity of the patients. The system can effectively classify disease severity with only a small amount of data, and achieve 96-99% accuracy using KNN, SVM, and GB classifiers. However, there are some issues the authors should address as follows:

  1. Is the study's classification of disease severity (levels I-IV) the same as the medical classification of Parkinson's disease? Who performs the labeling of level I-IV? How many people do the labeling?
  2. This study is based on two videos on Youtube, namely the normal gait and the gait of Parkinson's patients. Does the classification of the linear combination match the actual severity of the patient?
  3. How does this study label its training data? Does it label each gait one by one?
  4. The OpenCV algorithm used is not 100% accurate, especially the skeleton detection on the side is more inaccurate. Do you want to shoot from the front or from the side? If you shoot from the side, how to solve the problem of inaccurate skeleton detection?
  5. “The synthetic data generated was obtained by using a linear combination of the each data segmentation curve from both normal and Parkinsonian gait to simulate additional examples at different severity levels.” Is there any literature to synthesize different levels of PG in this way? Is there a way to verify the correctness of its theory? And will it be too little to train a machine learning model with only 2 video data? If there is a lack of training data of different heights, body shapes, and different levels of PG, will it be impossible to identify the severity of PG when new test subject data is entered later?
  6. How much data is generated in the final synthesis process?
  7. Please move the legend in Figure 11 outside the figure so as not to affect the reading of the histogram.
  8. What would happen if the real PG dataset from the literature was fed into the KNN, SVM, and GB models already trained in this study? This will help to confirm the practicality of this method.
  9. Some of the references are incorrectly formatted, please refer to the author guideline for revision.

Author Response

We sincerely appreciate the time and valuable suggestions from the reviewer. Our response letter and the revised version of the paper are attached.

Reviewer 2 Report

Thank you for the opportunity to review this manuscript on A Vision-based System for Stage Classification of Parkinsonian Gait Using Machine Learning and Synthetic Data. The main goal of this approach was that the artificially manufactured information can be used to expand on existing real data and train deep learning models. In this paper, a vision-based system that performs stage classification of Parkinson’s disease is proposed. 
Parkinson’s Disease being the second most hazardous neurological disorder has developed its roots in damaging people’s quality of life. The ineffectiveness of clinical rating scales makes the Parkinson’s Disease diagnosis a very complicated task. Thus, more efficient systems are required to perform an automated evaluation of Parkinson’s Disease for its earlier detection and to enhance life expectancy rate. Gait based clinical diagnosis can provide useful indications regarding the presence of Parkinson’s Disease. From recent years, computer vision-based analysis is in great demand and seems to be highly effective in Parkinson’s Disease inspection. 
The created idea is interesting, but I am not convinced by its reliability and application in clinical practice. I am afraid that it is too imprecise in relation to patients with Parkinson's. Please explain in more detail the advantages of the system used marker-based (model- build) compared to  marker-less (model-free).
Motion quantification thorough kinematic analysis can help understand the mechanisms underlying functional improvement following an intervention, and, in this sense, the paper under review is of scientific interest.
There is a need for simple, fast, easy-to-use, and applicable methods to allow routinely functional evaluation of patients with different pathologies and clinical conditions in neurology. The method presented in this paper is not described in terms of applicability, which is another concern.
I encourage the authors, who I believe made an extensive analysis, to discuss how results can help the rehabilitation decision-making process.

Author Response

We sincerely appreciate the time and valuable suggestion from the reviewer. Our response letter and the revised manuscript is attached.

Round 2

Reviewer 1 Report

This study proposes a visualization system to collect the gait of the patients with Parkinson's disease and analyze the disease severity of the patients. The system can effectively classify disease severity with only a small amount of data, and achieve 96-99% accuracy using KNN, SVM, and GB classifiers. Authors almost and completely answered all questions. However, there are some format mistakes needed to correct.
1. The words used in the article should be unified, ex: F1-score appears in the texts and figures has four formats of F1-score, F1 score, F1score, and F1.
2. The grading of PG severity, using healthy, mild, medium, and high in this manuscript. Fewer articles use the words of “medium” and “high”. I suggest to use "Healthy, mild, moderate, and severe" instead.
3. There are still many errors in the format of References, which the author must revise carefully. Ex: reference 3 should be “Hu, Q.; Tang, X.; Tang, W. A Real-Time Patient-Specific Sleeping Posture Recognition System Using Pressure Sensitive Conductive Sheet and Transfer Learning. IEEE Sensors Journal 2020, 21, 1– 11”.

Author Response

We sincerely appreciate the time and expertise of the reviewer. The paper has been edited according to the reviewer's suggestion.

Reviewer 2 Report

Thank you for the improved version of the manuscript, I am satisfied.

Author Response

We sincerely appreciate the time and expertise of the reviewer.